# STAG2 deficiency induces interferon responses via cGAS-STING pathway and restricts virus infection

Siyuan Ding[1,2,3], Jonathan Diep[1], Ningguo Feng[1,2,3], Lili Ren[1,2,3,4], Bin Li[5], Yaw Shin Ooi[1], Xin Wang[6,10], Kevin F. Brulois[3,7], Linda L. Yasukawa[1,2,3], Xingnan Li[8], Calvin J. Kuo[8], David A. Solomon [iD] [9], Jan E. Carette[1] & Harry B. Greenberg[1,2,3]

Cohesin is a multi-subunit nuclear protein complex that coordinates sister chromatid separation during cell division. Highly frequent somatic mutations in genes encoding core cohesin subunits have been reported in multiple cancer types. Here, using a genome-wide CRISPR-Cas9 screening approach to identify host dependency factors and novel innate immune regulators of rotavirus (RV) infection, we demonstrate that the loss of *STAG2*, an important component of the cohesin complex, confers resistance to RV replication in cell culture and human intestinal enteroids. Mechanistically, *STAG2* deficiency results in spontaneous genomic DNA damage and robust interferon (IFN) expression via the cGAS-STING cytosolic DNA-sensing pathway. The resultant activation of JAK-STAT signaling and IFN-stimulated gene (ISG) expression broadly protects against virus infections, including RVs. Our work highlights a previously undocumented role of the cohesin complex in regulating IFN homeostasis and identifies new therapeutic avenues for manipulating the innate immunity.

[1] Department of Microbiology and Immunology, Stanford University, Stanford, CA 94305, USA. [2] Department of Medicine, Division of Gastroenterology and Hepatology, Stanford University, Stanford, CA 94305, USA. [3] Palo Alto Veterans Institute of Research, VA Palo Alto Health Care System, Palo Alto, CA 94304, USA. [4] School of Pharmaceutical Sciences, Nanjing Tech University, 211816 Nanjing, China. [5] Institute of Veterinary Medicine, Jiangsu Academy of Agricultural Sciences, 210014 Nanjing, China. [6] Department of Immunology, Cleveland Clinic, Cleveland, OH 44195, USA. [7] Department of Pathology, Stanford University, Stanford, CA 94305, USA. [8] Department of Medicine, Division of Hematology, Stanford University, Stanford, CA 94305, USA. [9] Department of Pathology, University of California, San Francisco, CA 94143, USA. [10] Key Laboratory of Marine Drugs, Ministry of Education, Ocean University of China, 266071 Qingdao, China. Correspondence and requests for materials should be addressed to H.B.G. (email: hbgreen@stanford.edu)

Genome-wide CRISPR-Cas9 loss-of-function screens have emerged as a powerful tool to interrogate pathogen–host interaction at the molecular level[1]. This new method enables complete disruption of target genes and thereby identifies high-confidence host protein candidates that are critical for pathogen replication[1]. Novel host factors for several viral pathogens, including dengue virus, Zika virus, West Nile virus, hepatitis C virus, HIV, and murine norovirus, have been recently uncovered using this approach[2–6]. Rotaviruses (RVs) are icosahedral viruses with segmented, double-stranded RNA genomes[7]. Clinically, RVs are a leading cause of severe gastroenteritis and diarrheal mortality in young children worldwide[8], causing over 200,000 deaths annually. In addition to their public health relevance, RVs serve as a prototypic enteric model system to investigate host innate immune responses to microbial pathogens in the intestinal mucosa. For instance, we have recently identified the type I and type III interferons (IFNs) as key determinants of RV host range restriction[9]. We also recently found the intestine-specific Nlrp9b inflammasome to be a cardinal host factor that protects against RV infection in vivo[10]. Despite recent advances in proteomics and small interfering RNA (siRNA)-based screens for RVs[11–14], the nature and identity of many pro-RV and anti-RV host factors remain unknown.

Here we employ a genome-scale CRISPR-Cas9 screening approach to systematically identify host factors that support RV replication as well as novel regulators of the host innate immune signaling. We uncover several uncharacterized cellular pathways and stromal antigen 2 (encoded by *STAG2*) that facilitate RV infection. Importantly, depletion of *STAG2* triggers host genomic DNA damage, recognition of cytoplasmic microchromatin, and the activation of cGAS-STING-IRF3 signaling, which culminates in IFN production and resistance to multiple RNA virus infections.

## Results

**Genetic screen identifies novel pro-rotaviral host factors.** To enable the genome-wide CRISPR-Cas9 screen for RV host dependency factors, we first transduced H1-Hela cells with a pool of lentiviruses encoding Cas9 and the GeCKO single-guide RNA library (sgRNA, 6 per coding gene, 4 per miRNA locus, and 2000 non-targeting controls) as described[3]. This heterogeneous H1-Hela cell population was exposed to the cytopathic NCDV strain of bovine RV (G6, P[1]) for multiple rounds of infection until the appearance of visibly apparent survival colonies, which were then harvested and processed for next-generation sequencing (Fig. 1a). Ranking the enriched genes using MAGeCK algorithm[15] revealed a large panel of novel host-dependency factors for RV infection (Fig. 1b). Using this screening strategy, consistent with published studies[12,14,16], we identified several genes known to be critical for RV infection (Supplementary Data 1), including *SLC35A1*, *GNE*, and *CMAS* in the sialic acid synthesis pathway; *UGCG*, which catalyzes glycosphingolipid biosynthesis; and *FA2H*, a fatty acid 2-hydroxylase. We also found *LATS2* (hit #39), a Hippo pathway kinase recently shown to negatively regulate IFN activity[17], highlighting the fact that our screen is able to uncover innate

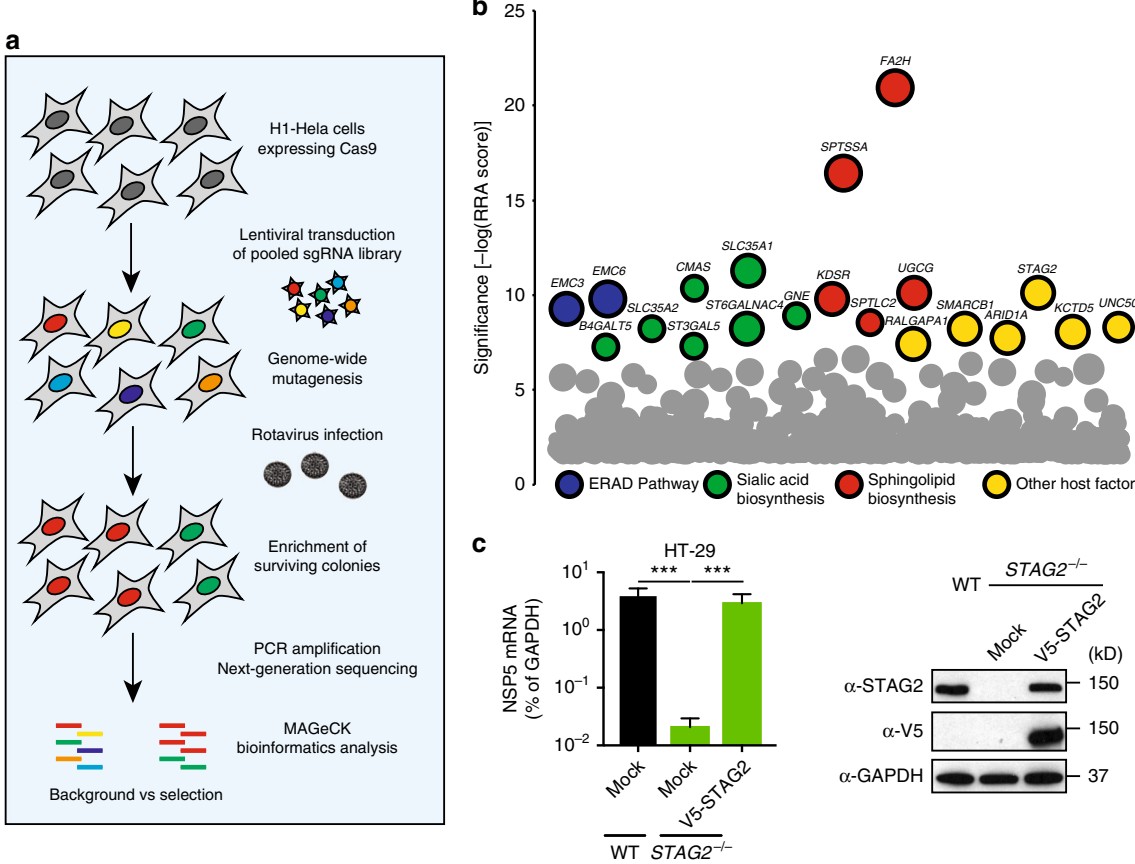

**Fig. 1** A genome-wide CRISPR-Cas9 screen reveals STAG2 as a pro-RV host factor. **a** Schematic flowchart for RV CRISPR-based loss-of-function screening approach. **b** Bubble plot of host factors essential to RV infection. The top 20 genes were colored and grouped by function. Size of bubbles corresponds to the number of significant sgRNAs scored for each gene. **c** Wild-type (WT), *STAG2*$^{-/-}$, and *STAG2*$^{-/-}$ HT-29 cells transduced with V5-tagged STAG2 were infected with RV (MOI = 1) and viral NSP5 mRNA level was measured at 24 h.p.i. by RT-qPCR (left panel). Cell lysates were analyzed by western blot with the indicated antibodies (right panel). For **c**, experiments were repeated at least three times in triplicates. Data are represented as mean ± SEM (***$P \leq 0.001$)

immunity-associated host factors. Notably, we identified *STAG2*, a core subunit of the nuclear cohesin complex[18,19], as a top candidate (#6) in the screen. In addition to *STAG2*, we noticed an enrichment of *SMC3* (#209) and *STAG1* (#610), two other cohesin components. It was particularly interesting that the replication of RV was highly restricted by the absence of several key components of the nuclear cohesin complex, even though RV replication is thought to take place exclusively in the cytoplasm[7].

To further examine this fascinating phenotype, we first validated the screen results by knocking out *STAG2* in Caco-2 and T84 cells, both human colonic cancer-derived epithelial cell lines (Supplementary Fig. 1A). Consistent with our findings in H1-Hela cells, RV replication was significantly reduced in these two STAG2-depleted cell lines (Supplementary Fig. 1B). To study the role of STAG2 in RV infection more directly, we then generated a single clonal STAG2 knockout in HT-29 cells, another human intestinal epithelial cell (IEC) line commonly used for RV studies. Complete STAG2 deletion was confirmed by both western blot and Sanger sequencing (Supplementary Fig. 1C). These cells did not exhibit severe defects in survival or proliferation (Supplementary Figs. 1D-F). Importantly, RV infectivity was significantly decreased (~3 log) in $STAG2^{-/-}$ HT-29 cells compared to wild-type (WT) cells, as measured by the expression of the gene encoding RV NSP5 (Fig. 1c). Surprisingly, we found a profound defect in multiple steps of

the RV replication cycle, including transcription, viroplasm formation, and release of virus progeny into the supernatant (Supplementary Figs. 2A-D). Susceptibility to RV infection was restored upon exogenous expression of WT STAG2 in $STAG2^{-/-}$ HT-29 cells (Fig. 1c), suggesting that the effect was specifically due to the loss of STAG2. Besides the bovine RV NCDV strain, we tested additional human and animal RV strains and they were all reduced in the absence of STAG2 (Supplementary Fig. 3A).

**Loss of *STAG2* activates IFN and ISG expression**. Remarkably, the replication of several unrelated RNA viruses, including vesicular stomatitis virus (VSV), chikungunya virus (CHIKV), and two subtypes of influenza A virus, was also significantly inhibited in $STAG2^{-/-}$ HT-29 cells compared to WT cells (Fig. 2a and Supplementary Fig. 3B). Conversely, the replication of several flaviviruses and DNA viruses was not affected (Supplementary Fig. 3C). Taken together, these data suggest that the loss of STAG2 likely leads to an alteration of signaling pathways within host cells that is commonly shared by RV, VSV, CHIKV, and influenza viruses.

To identify the mechanism by which the loss of STAG2 leads to a suppression of RV growth, we first performed an unbiased RNA-sequencing analysis, using two different platforms, to profile the transcriptome of WT and $STAG2^{-/-}$ HT-29 cells.

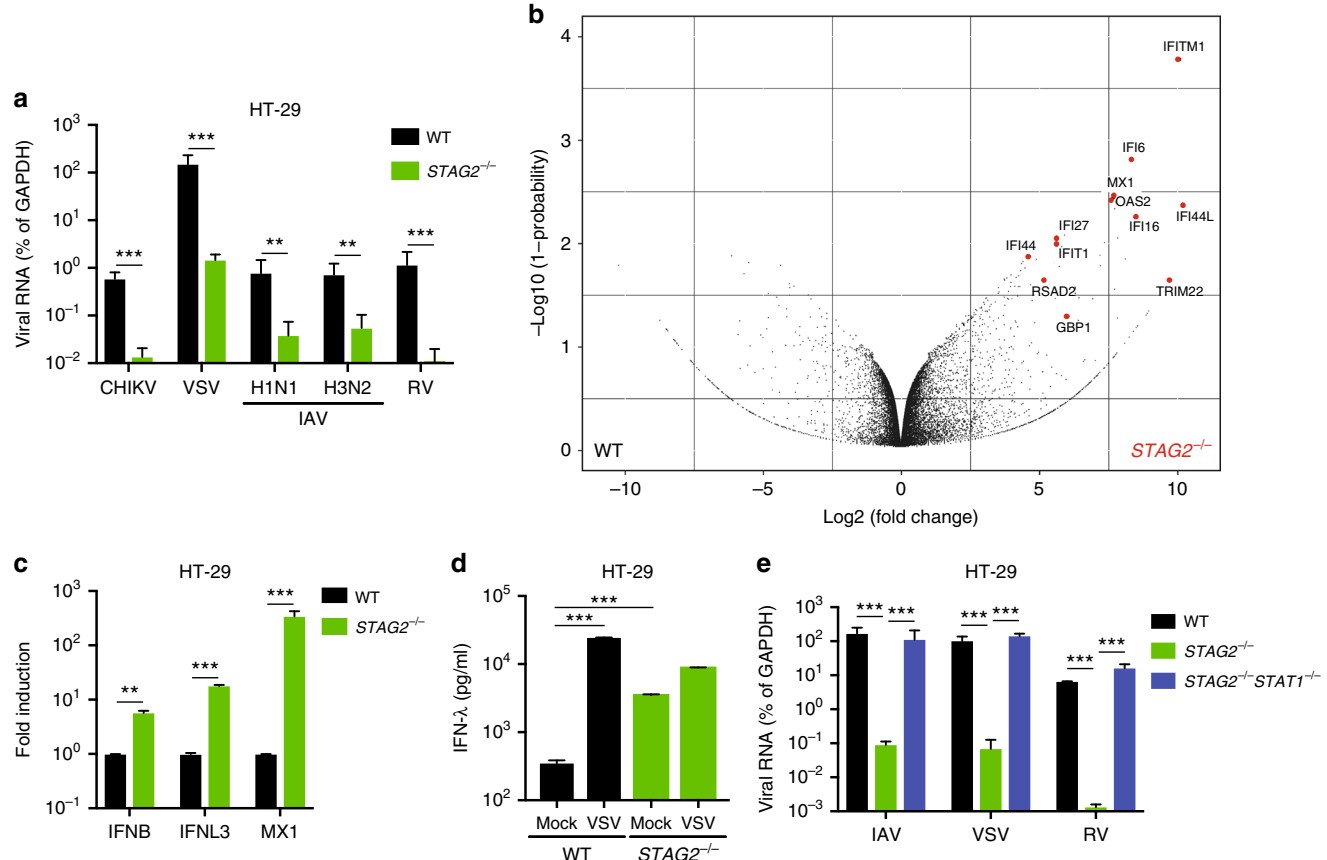

**Fig. 2** Loss of STAG2 triggers excessive IFN production. **a** WT and $STAG2^{-/-}$ HT-29 cells were infected with the indicated viruses (MOI = 1) and respective viral genes were measured by RT-qPCR at 24 h.p.i. CHIKV chikungunya virus, VSV vesicular stomatitis virus, IAV influenza A virus. **b** Volcano plot of RNA-sequencing data (BGI system) from uninfected WT and $STAG2^{-/-}$ HT-29 cells. **c** IFN and ISG expression was examined by RT-qPCR. **d** IFN-λ secretion was measured by ELISA. **e** WT, $STAG2^{-/-}$, and $STAG2^{-/-}STAT1^{-/-}$ HT-29 cells were infected with IAV, VSV, and RV (MOI = 1) and viral mRNA level was measured at 24 h.p.i. by RT-qPCR. For all figures except **b**, experiments were repeated at least three times in triplicates. Experiment of **b** was performed in duplicates on two different sequencing platforms. Data are represented as mean ± SEM (**p ≤ 0.01; ***p ≤ 0.001)

Strikingly, canonical ISGs such as MX1, IFITM1, and IFI6 were significantly upregulated (>100-fold) in $STAG2^{-/-}$ HT-29 cells compared to their WT counterparts, in the absence of any virus infection (Fig. 2b and Supplementary Data 2). Gene ontology pathway analysis revealed a distinct IFN signature in the $STAG2^{-/-}$ HT-29 cells (Supplementary Fig. 4A). Transfer of conditioned media from $STAG2^{-/-}$ HT-29 cells elicited luciferase expression driven by IFN-stimulated responsive element promoters (Supplementary Fig. 4B). An increase in ISG expression and secretion of IFN-λ in the supernatant in $STAG2^{-/-}$ HT-29 cells were further demonstrated by reverse transcriptase quantitative PCR (RT-qPCR) and enzyme-linked immunosorbent assay (ELISA), respectively (Fig. 2c, d). In contrast, we did not detect an increase in the secretion of IFN-α/β and tumor necrosis factor (TNF)-α, which were found in VSV-infected WT HT-29 cells (Supplementary Fig. 4C). Finally, we found a partial nuclear localization of IRF3 in uninfected $STAG2^{-/-}$ HT-29 cells, indicative of spontaneous IFN activation, similar to that seen in VSV-infected cells (Supplementary Fig. 4D).

We hypothesized that the observed IFN activation mediates broad resistance to virus infection in STAG2-deficient cells. Consistent with this hypothesis, treatment with the selective Janus-activated kinase 1/2 (JAK1/2) inhibitor ruxolitinib, which effectively blocks all three types of IFN signaling[20], led to a dramatic reduction in ISG expression in $STAG2^{-/-}$ HT-29 cells (Supplementary Fig. 4E). Furthermore, CRISPR-induced knockout of signal transducer and activator of transcription factor 1 (STAT1) in the background of STAG2 deficiency completely abolished ISG expression and these double knockout cells regained their susceptibility to RV and VSV infections to levels comparable to WT HT-29 cells (Fig. 2e and Supplementary Fig. 4F), supporting the conclusion that a hyperactive IFN-JAK-

STAT pathway is responsible for suppressing multiple viral infections in $STAG2^{-/-}$ cells.

**STAG2 deletion triggers DNA damage and DNA-sensing pathway.** We next sought to determine mechanistically how the cell-intrinsic IFN activation occurred in the $STAG2^{-/-}$ cells. We assayed the phosphorylation status of several signaling pathways, based on cohesin's function in CTCF locus binding and cell division[18,21]. Interestingly, in the absence of ionizing or genotoxic agents, strong phosphorylation of the histone H2A variant H2AX (γH2AX), a hallmark of extensive DNA damage, was observed in $STAG2^{-/-}$ cells, concurrent with strong STAT1 phosphorylation (Fig. 3a). Further analysis suggested that this DNA damage response was mediated through double-stranded DNA breaks (DSBs) and the downstream ATM-53BP1 pathway (Supplementary Fig. 5A). Stronger 53BP1 staining was detected in $STAG2^{-/-}$ HT-29 cells than in WT cells while single-stranded DNA breaks, as measured by single-cell electrophoresis, were not visibly different between these two cell populations (Supplementary Fig. 5B). Coincidental with increased DSBs, we found markedly more micronuclei-like cytoplasmic DNA in the $STAG2^{-/-}$ cells and this increased DNA did not exclusively co-localize with the mitochondria (Fig. 3b). Cytosolic DNA is a well-characterized pathogen-associated molecular pattern (PAMP) that stimulates IFN production through the cGAS-STING-sensing pathway[22,23]. Paralleling the results of STAG2 deficiency, we found that etoposide treatment, which induced chromosome instability and increased γH2AX levels (Supplementary Fig. 5C), also activated STAT1 phosphorylation and ISG expression (Supplementary Figs. 5C and 5D). Importantly, RV replication was also significantly inhibited in etoposide-primed cells (Supplementary

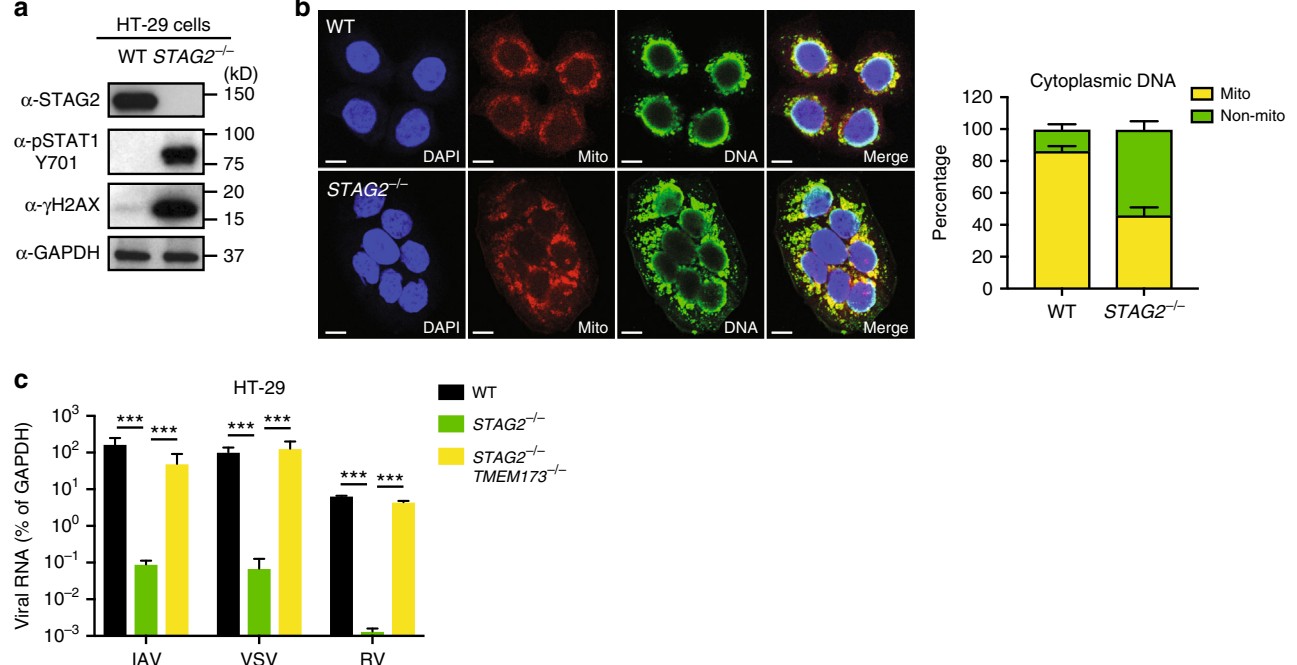

**Fig. 3** STAG2 deficiency mediates DNA damage and activation of the cGAS-STING signaling. **a** WT and $STAG2^{-/-}$ HT-29 cells were analyzed by western blot with the indicated antibodies. **b** Immunofluorescence analysis of WT and $STAG2^{-/-}$ HT-29 cells: nucleus (DAPI, blue), mitochondria (MitoTracker, red), and cytoplasmic DNA (green). In merged panels, mitochondrial DNA is shown in yellow and non-mitochondrial cytoplasmic DNA is shown in green. Scale bar: 8 μm. **c** WT, $STAG2^{-/-}$, and $STAG2^{-/-}STING^{-/-}$ HT-29 cells were infected with IAV, VSV, and RV (MOI = 1) and viral mRNA level was measured at 24 h.p.i. by RT-qPCR. IAV influenza A virus, VSV vesicular stomatitis virus. For all figures, experiments were repeated at least three times in triplicates. Data are represented as mean ± SEM (***p ≤ 0.001)

Fig. 5E), suggesting that the suppression is not restricted to STAG2 and is shared by various host DNA damage induction mechanisms.

To pinpoint the DNA sensor responsible for IFN induction in the $STAG2^{-/-}$ cells, we carried out a small-scale siRNA screen to knock down all reported DNA sensors and their downstream adaptor proteins. Our results indicated that siRNA silencing of either cGAS or STING led to a significant decrease in IFN expression in $STAG2^{-/-}$ HT-29 cells (Supplementary Fig. 6A), whereas knockdown of MAVS had a minimal effect (Supplementary Fig. 6B). A combined siRNA treatment targeting both cGAS and STING resulted in the greatest level of IFN inhibition, similar to that observed with a dual silencing of IRF3 and β-TrCP (Supplementary Fig. 6A), two adaptor proteins downstream of the cGAS-STING-TBK1/nuclear factor-κB pathway[24].

To further examine the role of STING in IFN induction in the $STAG2^{-/-}$ cells, we generated WT and $STAG2^{-/-}$ HT-29 cell lines stably expressing HA-STING to circumvent a lack of antibodies suitable for direct immunofluorescence analysis of STING. As expected, STING exhibited an activation status, marked by perinuclear, punctate structures, in $STAG2^{-/-}$ HT-29 cells, as opposed to the diffuse pattern in WT cells (Supplementary Fig. 7A). We then constructed double knockout HT-29 cells that lack both STAG2 and TMEM173, the gene that encodes STING (Supplementary Fig. 7B). These double knockout cells were completely unresponsive to cytosolic DNA stimuli (Supplementary Fig. 7C), and STAT1 hyper-phosphorylation was abolished compared to STAG2 single knockout cells (Supplementary Fig. 7B). Most importantly, they phenocopied our prior results in $STAG2^{-/-}STAT1^{-/-}$ cells and restored infectivity for RNA viruses (Fig. 3c). We also noticed a decrease in DSBs in $STAG2^{-/-}STING^{-/-}$ cells (Supplementary Fig. 7B), consistent

with the recent observation that IFN signaling potentiates an ATM-dependent DNA damage response[25].

In attempting to validate these findings in a second cell line, we made the initially perplexing observation that HEK293 cells, a human fibroblast cell line, upon clonal STAG2 deletion, did not have an abnormal IFN signature despite high levels of DNA damage induction (Supplementary Fig. 8A). Nor were these $STAG2^{-/-}$ HEK293 cells characterized by an IFN signature and lower levels of RV replication compared to WT HEK293 cells (Supplementary Fig. 8B). Coincidentally, we found that the HEK293 cells were profoundly defective in their cytosolic DNA-sensing IFN induction capability (Supplementary Fig. 8C). We validated that several HEK293 cell lines, regardless of their sources, naturally lack cGAS or STING expression at the protein level (Supplementary Fig. 8C). Therefore, to further explore the role of cGAS-STING pathway, we established an HEK293 cell line that stably express Flag-tagged cGAS and HA-tagged STING (Supplementary Fig. 8D). The restoration of an intact cytosolic DNA-sensing pathway was confirmed by adenovirus infection, which did not trigger IFN production in WT HEK293 cells (Supplementary Fig. 8E). As predicted, we observed a potent IFN induction and STAT1 phosphorylation in this new cell line upon genetic depletion of STAG2 (Supplementary Fig. 8D). In addition, we reconstituted the DNA-sensing-competent $STAG2^{-/-}$ HEK293 cells with STAG2 deletion mutants and examined their respective ability to inhibit DDR signaling and IFN induction (Supplementary Fig. 9A). The full-length WT STAG2 but none of the truncation mutants rescued aberrant DSB-ATM signaling and the subsequent STAT1 activation (Supplementary Fig. 9B). These data suggest that the STAG2 mutations that compromise the integrity of the cohesin complex[26] may contribute to the induction of chromosomal instability and IFN dysregulation.

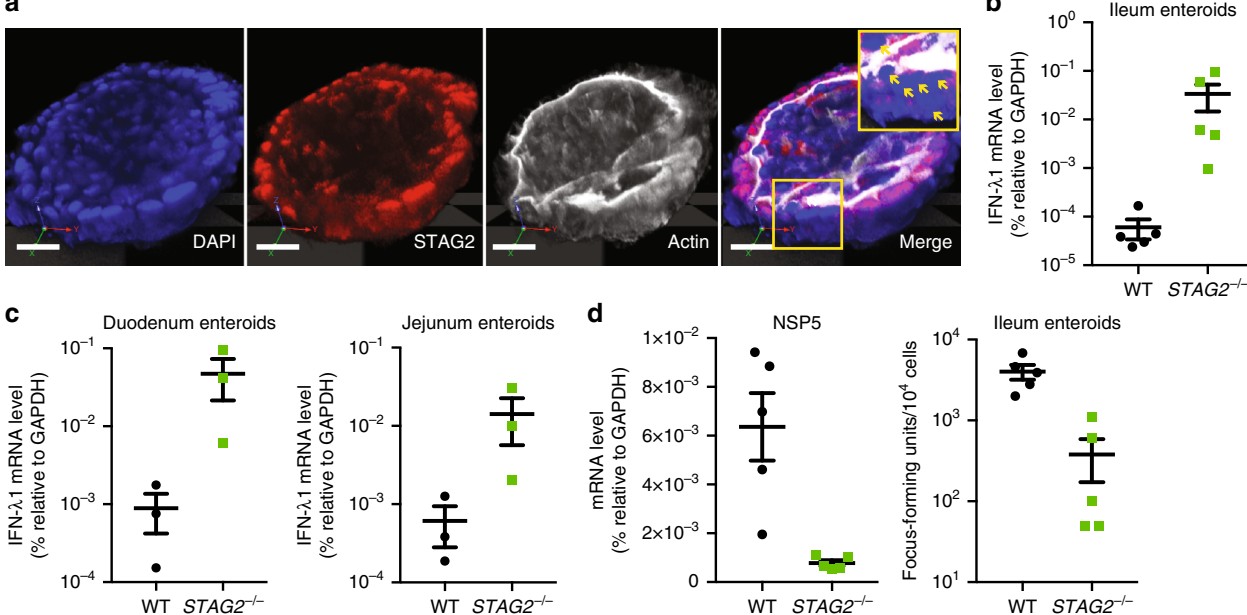

**Fig. 4** STAG2 depletion induces IFN expression in human intestinal enteroids. **a** 3D immunofluorescence analysis of a single ileal enteroid transduced with a lentiviral vector expressing Cas9 and STAG2 sgRNA: nucleus (DAPI, blue), STAG2 (red), and actin (phalloidin, white). Scale bar: 50 μm. Yellow box marks the region enlarged in the inset. Yellow arrows in the inset panel indicate IECs that are completely STAG2 knocked out. **b**, **c** Steady-state IFN-λ1 expression was measured by RT-qPCR in WT and partial $STAG2^{-/-}$ ileum enteroids ($n = 4$) (**b**) and in duodenum and jejunum enteroids ($n = 3$) (**c**). **d** WT and partial $STAG2^{-/-}$ enteroids were infected with RV (MOI = 5) and viral NSP5 level and virus yields were determined at 24 h.p.i. All scatter plots in **b**–**d** are calculated based on comparison between WT and STAG2 mosaic (~30% KO) enteroids. For all figures, experiments were repeated at least three times in triplicates. Data are represented as mean ± SEM

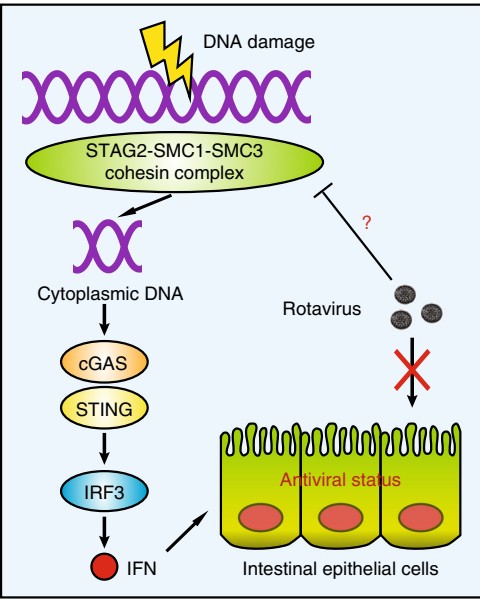

**Fig. 5** Working model of RV–cohesin interaction. Host genomic DNA damage induced by cohesin deficiency led to an increase in the levels of cytoplasmic DNA, which feeds into the cGAS-STING DNA-sensing pathway to activate IFN and ISG expression. These processes enable the host cells to enter an antiviral status and render them resistant to rotavirus infection. Future studies will focus on whether rotavirus has evolved strategies to dampen the host DNA damage response and subsequent IFN production. Cohesin is multimeric nuclear protein complex that includes STAG2 and is associated with vital roles during cell division. Here, in a genome-wide CRISPR-Cas9 screen, the authors identify a novel role of STAG2 as a crucial component of the innate immune response to rotavirus

**STAG2 depletion reduces RV infectivity in human enteroids.**
Finally, we extended our findings in transformed cell lines to more physiologically relevant human intestinal enteroids, a primary human small bowel IEC culture that recapitulates enteric epithelial cell diversity and supports RV infection[27]. We partially depleted STAG2 expression in ileum enteroids derived from five healthy individuals (Fig. 4a). A complete knockout of STAG2 at the protein level was detected in approximately 30% of the primary IECs (white arrows in Fig. 4a) and it was sufficient to induce a hyperactive type III IFN response in these enteroids (Fig. 4b), even greater than that observed in $STAG2^{-/-}$ HT-29 cells (Fig. 2c). Our results were further confirmed in additional experiments using duodenum and jejunum enteroids (Fig. 4c), suggesting that the role of STAG2 in tightly regulating IFN activity is conserved in various sections of the human small intestine. Accordingly, RV replication was significantly reduced in STAG2 partially depleted human enteroids (Fig. 4d), as evidenced by the RV RNA levels and yield of infectious virus.

## Discussion
In this study, we report a critical function of cohesin complex in immune homeostasis by preventing host DNA from being recognized as a PAMP (Fig. 5). The genome-wide loss-of-function screen and RNA-seq dataset revealed that STAG2 depletion elicits an excessive IFN expression. The constitutive IFN activation is reminiscent of recent publications of $Trex1^{-/-}$, $Atm^{-/-}$, and $Lats1/2^{-/-}$ cells, where the basal expression levels of IFNs were elevated and exhibited an autoimmune-like manifestation[17,28,29]. Given the association between cohesin and DNA damage[30,31], it is tempting to speculate that other proteins in this signaling pathway are likely also critical for modulating IFN

responses and preventing autoimmune- or lupus-like syndromes[32]. During the period of peer review of this manuscript, several studies were published in agreement with our finding that cytoplasmic DNA, originated from the host nucleus, is generated during senescence or genomic instability and activates cGAS-STING signaling and IFN induction[33–36]. Our findings may also have implications for cancer biology studies. STAG2 mutations are frequently detected in multiple tumor types and its loss of function is believed to induce aneuploidy[19,37]. Our examination of two glioblastoma cell lines (U138MG and H4 cells) that naturally do not express STAG2 indicates that both are defective in the cGAS-STING pathway (Supplementary Fig. 9C). By downregulation of the DNA sensor cGAS, the adaptor protein STING, or both, these STAG2-deficient tumors may benefit from the unchecked proliferation due to chromosomal multiplication while not having to face the potential adverse consequences of eliciting antitumoral IFN activation. Furthermore, our results imply that the current cGAS or STING ligands under development for treating or serving as adjuvants in certain cancers[38,39] will likely not work in tumor types defective in STAG2 and the cytosolic DNA-sensing pathway. Our data also imply that for the other cancers that harbor STAG2 mutations and an intact DNA-sensing mechanism, depending on the type of IFN induction, certain oncolytic viral vectors may also fail to function as expected.

## Methods
**Cells and reagents**. H1-Hela (CRL-1958), Caco-2 (HTB-37), T84 (CCL-248), HEK293 (CRL-1573), HEK293T (CRL-3216), U138 (HTB-16), and H4 (HTB-148) cells were obtained from American Type Culture Collection (ATCC) and cultured in complete Dulbecco's modified Eagle's medium (DMEM) medium; HT-29 (HTB38) cells were obtained from ATCC and cultured in complete advanced DMEM/F12 medium; MA104 (CRL-2378.1) cells were obtained from ATCC and cultured in complete M199 medium. $STAG2^{-/-}$ HT-29 cells rescued with pLX304-STAG2 were cultured in the presence of blasticidin (5 μg/ml). WT or $STAG2^{-/-}$ HT-29 cells expressing HA-STING were generated using puromycin selection (1 μg/ml). WT or $STAG2^{-/-}$ HEK293 cells expressing HA-STING and Flag-cGAS were selected under puromycin (1 μg/ml) and G418 (500 μg/ml). Full-length V5-tagged STAG2 in pLX304 vector was purchased from DNASU Plasmid Repository (HsCD00438827). Flag-SBP tagged STAG2 full-length plasmid and mutants (S97X, S653X, S1075X, Q1117X, S1215X) in pcDNA3.1 vector were purchased from Addgene (#73963, 73964, 73965, 73967, 73968, 73970, respectively). pSpCas9(BB)-2A-GFP (PX458) and lenti-CRISPR_v2 were purchased from Addgene (#48138 and 52961, respectively). pCMV6-Entry-Flag-cGAS was purchased from Origene (RC212386). pcDNA3.1-HA-STING was previously described[40]. Ruxolitinib was reconstituted at 10 mM stock solution in dimethyl sulfoxide and used at 100 nM in cell culture (Selleckchem, S1378).

**Genome-wide CRISPR-Cas9 screen**. The GeCKO v2.0 Human CRISPR Knockout Pooled Library from MIT Zhang lab was used to generate heterogeneous H1-Hela knockout cell population as previously described[3]. A total of $2.8 \times 10^8$ mutagenized cells ($1.4 \times 10^8$ cells for both library A and B) were infected with bovine RV NCDV strain at a multiplicity of infection (MOI) = 10 in serum-free medium for 48 h, recovered in complete DMEM for 24 h, and repeated for an additional 9 rounds of infection until the appearance of visibly viable colonies. Genomic DNA was harvested from the live cells and sgRNAs were amplified for sequencing on Illumina NextSeq platform. The MAGeCK algorithm[15] was used for data analysis, taking into account multiple different sgRNAs per gene, number of sequencing reads per gene, and the enrichment of sgRNAs compared to the uninfected pooled library. A complete list of genes and scores can be found in Supplementary Data 1.

**CRISPR-Cas9 knockout cells**. Single clonal knockout HT-29 and HEK293 cells were obtained using the PX458 vector that expresses Cas9 and sgRNA against STAG2, STAT1, and STING (Supplementary Table 1). Green fluorescent protein (GFP)-positive single cells were sorted at 48 h post-transfection using BD Aria II into 96-well plates (see Supplemental Information for gating strategy) and screened for knockout based on western blot and Sanger sequencing. Pooled knockout Caco-2, T84 cells, and human intestinal enteroids were obtained by lentiviral transduction with the lenti-CRISPR_v2 vector that expresses Cas9 and STAG2 sgRNA for a minimum of 14 days under puromycin selection.

**Viruses and virus infections**. All human and animal RV strains used in this study were propagated in MA104 cells and RV infection was performed as previously described[11]. Recombinant VSV (strain Indiana) expressing GFP was a kind gift from Dr. Jack Rose (Yale University). Influenza A viruses (H1N1 A/California/7/2009, H3N2 A/Victoria/361/2011), enterovirus-D68 (strain US/MO/14-18947), human rhinovirus (strain A2), ZIKV (strain P6-740), DENV-1 (strain 276RKI), CHIKV (strain 181/25), and adenovirus serotype 5 were used as previously described[13]. Vaccinia virus (strain MVA) and SV40 (strain EK) were purchased from ATCC.

Lentiviruses used in this study include pLX304 vector encoding V5-tagged STAG2 and lenti-CRISPR_v2 vector encoding Cas9 and STAG2 sgRNA. Both were packaged in HEK293T cells by co-transfection with psPAX2 and pMD2.G as previously described[41]. Supernatants were collected at 48 and 72 h post-transfection, passed through a 450 nm filter, and added to target cells in the presence of polybrene (8 µg/ml).

**Cell survival and proliferation**. The cell numbers of WT, $STAG2^{-/-}$ or $STAG2^{-/-}STING^{-/-}$ HT-29 cells in 24-well plates were determined by the luminescence production using the RealTime-Glo™ MT Cell Viability Assay (Promega, G9711) according to the manufacturer's instructions. WT and KO HT-29 cells in six-well plates were harvested and stained with the Live/Dead Fixable Aqua Dead Cell Stain Kit (Invitrogen, L34957) using flow cytometry.

**Single-cell electrophoresis**. WT or $STAG2^{-/-}$ HT-29 cells in 6-well plates were treated with 100 µM hydrogen peroxide at 4 °C for 20 min and harvested. Cells were then embedded into low melting point agarose for comet SCGE assay kit (Enzo Life Sciences, ADI-900-166) according to the manufacturer's instructions.

**Transfection**. HT-29 cells in 24-well plates were transfected with siRNA (2.5 µl of 5 µM) using DharmaFECT1 (1 µl per reaction) reagent (GE Life Sciences). All siRNAs used in this study were SMARTpool ON-TARGET siRNA purchased from Dharmacon (Supplementary Table 1). Canonical cGAMP (2'3' cGAMP) (8 µg/ml) was transfected into HT-29 and HEK293 cells using lipofectamine 2000 as previously described[42]. 2'2' cGAMP (Invivogen) was used as a negative control for which no IFN induction was observed.

**Western blot**. Cell lysates were harvested in RIPA buffer (Sigma-Aldrich) supplemented with protease inhibitor cocktail and phosphatase inhibitors (Roche). Sodium dodecyl sulfate-polyacrylamide gel electrophoresis was performed as previously described[43] using the following primary antibodies: p-ATM Ser1981 (CST, D6H9, #5883, 1:1000), Flag (Sigma, M2, #F3165, 1:1000), glyceraldehyde 3-phosphate dehydrogenase (BioLegend, #631402, 1:1000), γH2AX Ser139 (CST, #2577, 1:1000), MX1 (CST, D3W7I, #37849, 1:1000), STAG2 (CST, #4239, 1:1000), p-STAT1 Tyr 701 (CST, 58D6, #9167, 1:1000), STING (CST, D2P2F, #13647 S, 1:1000), and V5 (CST, D3H8Q, #13202, 1:1000). Secondary incubation was performed with anti-rabbit (CST, #7074, 1:5000) or anti-mouse (CST, #7076, 1:5000) immunoglobulin G horseradish peroxidase-linked antibodies. Protein bands were visualized with Clarity ECL substrate (Biorad, #170–5061), Amersham Hyperfilm (GE Healthcare), and STRUCTURIX X-ray film processor (GE Healthcare).

**ELISA for IFN and viral antigen**. Supernatants from mock or VSV-infected HT-29 cells were collected and measured by human IL-29/IL-28B (IFN-lambda 1/3) Duoset ELISA kit (R&D Systems, DY 1598B-05), human pan IFN-α ELISA kit (R&D, 41100-1), human IFN-β ELISA kit (R&D, 41410-1), and human TNF-α ELISA kit (R&D, DY210-05) according to the manufacturers' instructions.

**RNA extraction and RT-qPCR**. Total RNA was extracted as previously described[41]. Except for strand-specific PCR for NSP3, random hexamer was used for reverse transcription reaction. qPCR was performed with the Stratagene Mx3005P (Agilent) with a 25 µl reaction consisting of 50 ng of cDNA, 12.5 µl of Power SYBR Green master mix (Applied Biosystems), and 200 nM both forward and reverse primers. All SYBR Green primers (Supplementary Table 1) have been validated with dissociation curves and electrophoresis of the correct amplicon size.

**RNA sequencing**. Total RNA from WT and $STAG2^{-/-}$ HT-29 cells was extracted using the RNeasy Mini Kit (Qiagen). RNA sample quality was examined by NanoDrop spectrophotometer (Thermo Fisher) and Bioanalyzer 2100 (Agilent). Libraries were sequenced on both Illumina HiSeq4000 and BGISEQ-500 platforms. The SE reads were aligned to the hg19 build using Bowtie2 to map clean reads to reference gene and using HISAT2 to reference genome with the following parameters: --phred64 --sensitive -I 1 -X 1000. Reads were counted using Subread and differential gene expression analysis of BGI data was performed using DESeq2 (Supplementary Data 2).

**Immunofluorescence**. Confocal analysis was performed as previously described[13]. In brief, cells were fixed with 4% paraformaldehyde and stained with the following primary antibodies or fluorescent dyes: 53BP1 (Abcam, #ab21083, 1:200), dsDNA (Santa Cruz, HYB331-01, sc-58749, 1:100), HA (6E2)-Alexa Fluor 647 (CST, #3444, 1:100), p-IRF3 Ser396 (CST, D6O1M, #29047, 1:200), NSP2 (clone 191, 1:200)-Alexa Fluor 488, STAG2 (CST, D25A4, #5882), VP6 (clone 1E11, 1:200)-Alexa Fluor 488, MitoTracker Red CMXRos (Thermo Fisher, M7512), and Phalloidin-Alexa Fluor 647 (Thermo Fisher, A22287). Stained cells were washed with phosphate-buffered saline, mounted with Antifade Mountant with 4,6-diamidino-2-phenylindole (Thermo, P36962), and imaged with Zeiss LSM 710 Confocal Microscope. Z-stack was applied for imaging 3D human enteroids. Co-localization was analyzed by Volocity v5.2 (PerkinElmer) and 53BP1 punta were quantified with ImageJ.

**Human intestinal enteroids**. $Lgr5^+$ crypt cells were isolated from ileum biopsies of healthy individuals and small intestinal enteroids were cultured and infected as previously described[27]. In brief, after removal of Matrigel, enteroids were transduced with empty vector or lenti-CRISPR_v2 to deplete STAG2 for 10 days and then infected with human RV WI61 strain (MOI = 1) for 24 h. Mock or RV-infected enteroids were harvested for qPCR measuring RV NSP5 level and plaque assay for titration. All human intestinal enteroid experiments are in compliance with Stanford University human subject IRB requirements.

**Statistical analysis**. The bar graphs are displayed as means ± SEM. Statistical significance in Fig. 2a, c and in Supplementary Figs. 1B, 2D, 3A, 3C, 4E, 5A, 5D, and 6B was calculated by Student's t-test using Prism 7.0c (GraphPad) and indicated on each figure (*$p \leq 0.05$; **$p \leq 0.01$; ***$p \leq 0.001$). Statistical significance in Figs. 1c, 2e and 3c and in Supplementary Figs. 1E, 2A, 4B, 4C, 5E, 6A, 7C, and 8E was calculated by a pairwise analysis of variance (ANOVA) test using Prism 7.0c (GraphPad). Statistical significance of Supplementary Fig. 2B was calculated by two-way ANOVA test using Prism 7.0c (GraphPad). All experiments, unless otherwise noted, have been repeated at least three times.

**Data availability**. All relevant data are available from the paper or from the authors upon request.

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

## Acknowledgements

We would like to thank all members in the Greenberg lab for their support and discussion of the project. We thank Dr. Mrinmoy Sanyal for help in cell sorting to generate CRISPR knockout cells. We are grateful to Dr. John T. Patton (University of Maryland) and Dr. Terence S. Dermody (University of Pittsburgh) for discussion of the project. This work is supported by NIH grants R01 AI021362 and R56 AI021362 and by VA Merit review grant GRH0022 awarded to H.B.G. S.D. is supported by a Walter V. and Idun Berry Postdoctoral Fellowship, a Stanford Institute for Immunity, Transplantation and Infection (ITI) Young Investigator Award, and the Early Career Award from the Thrasher Research Fund. H.B.G., X.L., and C.J.K. are supported by NIH U19 AI116484. D.A.S. is supported by NIH DP5 OD021403. J.E.C. is supported by NIH DP2 AI104557.

## Author contributions

Conceptualization: S.D., J.D., X.W., J.E.C., and H.B.G.; investigation: S.D., J.D., N.F., L.R., B.L., Y.S.O., and L.L.Y.; formal analysis: K.F.B.; writing—original draft: S.D. and H.B.G.; writing—review and editing, S.D., J.D., B.L., K.F.B., C.J.K., and H.B.G.; funding acquisition: C.J.K. and H.B.G.; resources: X.L., C.J.K., and D.A.S.; supervision: J.E.C. and H.B.G.

## Additional information

**Competing interests:** The authors declare no competing interests.

