## [Peer Review File · Nature Communications]

Reviewers' comments:

Reviewer #1 (Remarks to the Author):

Review of Ding et al

This elegant and well-written paper by the Greenberg and Carette laboratories performed a genome-wide CRISPR screen to identify host factors required for rotavirus (RV) infection. While they identified a number of genes known to be critical for RV infection, they also identified STAG2, a component of the cohesion complex, as a 'hit' whereby its gene editing/deletion resulted in diminished RV infection as well reduction of infection of a number of related and unrelated RNA viruses. Through a series of well-designed, well-controlled, and detailed studies, they identified the mechanism of action: a deficiency of STAG2 results in spontaneous DNA damage, accumulation of DNA in the cytosol, activation of the cGAS-STING pathway, and spontaneous production of type I IFNs. This work, which was initially designed to identify RV-dependent host factors, instead defines a new regulator of IFN homeostasis and host pathway (cohesion complex members) that could have implications for autoimmune diseases (e.g., lupus, interferonopathies), cancer, and cancer therapies that modulate cell-intrinsic immunity. The strengths of the manuscript include its approach, mechanistic depth, novelty of findings, and clarity of the data. This will be an important contribution to the field of immunity and autoimmunity. There are only few minor suggestions for improvement.

Minor Criticisms.

1. Given that a deficiency of STAG2 results in spontaneous STING activation and IFN production, is it surprising that some but not all viruses are inhibited. In particular, DNA viruses, flaviviruses, and picornaviruses are not inhibited whereas rotavirus, alphaviruses (CHIKV), influenza, and VSV are inhibited. As this seems puzzling, the authors need to address this at least in the Discussion. What could be the explanation for why a robust IFN signature would inhibit some viruses but not others, as it seems likely that IFN pre-treatment would inhibit infection of all of these viruses. Related to this, in Figure 2, the authors show ISG and IFN- α induction by RT-PCR and IFN- α accumulation by ELISA in STAG2^{-/-} cells. Did they also directly measure secreted levels of IFN- α by ELISA using commercially available kits? Could a difference in induction levels of type I and III IFNs explain the disparity in effects against the different viruses?

2. In all main and Supplemental Figure legends, the authors should indicate clearly the number of independent biological experiments, the number of technical replicates, and the statistical test used to determine significance. Related to this point, in the Methods, the authors state they used a

Student's t test for all comparisons. However, the data in Fig 1c, 2e, 3c, S2a, S6a, and S7c requires an ANOVA with a multiple comparison's correction, as they compare across multiple groups.

3. Summary. The authors started the Summary with a few sentences on the cohesion complex. Given why the screen was performed (to identify host factors required for RV infection), they should re-order the text to reflect the true sequence of events in the paper. The statement "whether cohesion participates in regulation of innate immune responses..." seems disingenuous to this Reviewer – prior to this study, this was not even a consideration.

4. Human mutations in STAG2. Are there known mutations in STAG2 that have been detected in cancer patients (would seem so based on references 3 and 32)? Do these have any association with autoimmune phenotypes? Did the authors test any of the point mutations for loss-of-function vis a vis regulation of IFN responses?

5. Figure S4c. Why is the IRF3 staining so dim in the mock-treated WT cells relative to the mock-treated STAG2^{-/-} cells? Usually, IRF3 is expressed strongly at the basal level. Was it one of the induced genes in these cells by their RNAseq analysis?

6. The authors state a nuclear source of the DSB and cytosolic DNA in STAG2^{-/-} cells. How did they rule out cleaved mitochondrial DNA as a source?

Other Suggestions.

1. Line 127. Should read "secretion of IFN-□".

2. Line 182 and 186. The authors state that STAT1 hyper-phosphorylation was abolished and that there was a decrease in DSBs in STAG2^{-/-}STING^{-/-} cells in Fig S7b. This Figure panel does not show either. Is the data elsewhere?

3. Line 198. Perhaps the authors should state "naturally lack cGAS or STING expression"? See Lane 3 of Fig S8c.

4. Line 218 and Fig 4a. The authors state there was a complete KO of STAG2 in 30% of the primary IECs. Yet in the Figure (panels b-d), they refer to this as a complete KO. This seems confusing. Related to this, in the legend they state that “bar graphs in b-d are calculated based on comparison between WT and STAG2 mosaics. What bar graphs? (scatter plots are shown). Did the authors normalize the data in some way – this section needs to be clarified.

5. Lines 247-250. This conclusion seems obvious. cGAS and STING ligands will not work in cells lacking these pathways. Perhaps the authors can articulate a more interesting coda to this elegant paper?

Reviewer #2 (Remarks to the Author):

STAG2 deficiency induces IFN responses via cGAS STING pathway and restricts virus infection

By SY Ding et al (Corresponding author: HB Greenberg)

Submitted to Nat Commun (Editorial No. NCOMMS-17-19147-T)

General Comments

This manuscript describes the application of a recently established genome-wide CRISPR/Cas9 screening procedure to the identification of cellular genes important for the replication of species A rotaviruses (RVAs) in different gut-derived human cell lines and also in human gut stem cell-derived intestinal 3D (enteroid) cultures. Genes known to be critical for RVA replication were confirmed, e.g. genes involved in sialic acid and glycosphingolipid biosynthesis, and the gene encoding a fatty acid 2-hydroxylase. In addition the very interesting and novel observation was made, consisting of showing that suppression of the cellular cohesin gene STAG2, encoding stromal antigen 2, or of other genes of the cohesin complex confers significant decrease of RVA replication. This was shown in different cell lines including enteroids and with different RVA strains. The finding is also interesting since the cohesin complex functions in the nucleus while RVA replication takes place in the cytoplasm. The causative role of cohesin genes was supported by exogenous expression of STAG2 in STAG2^{-/-} cells which restored the ability of RVA to replicate. The replication of several other RNA viruses, such as vesicular stomatitis virus, chikungunya virus and different influenza A virus subtypes, was also inhibited in STAG2^{-/-} cells, but not of some DNA viruses. It turned out that in uninfected STAG2^{-/-} cells interferon (IFN) stimulatory genes (ISGs) such as encoding MX1, IFITM1, IFI6, etc were significantly upregulated, explaining the inhibitory effect on RNA virus replication. Mechanistically, it was found that STAG2^{-/-} cells exhibit signs of intracellular DNA damage, leading to activation of the

cGAS-STING pathway of DNA sensing and subsequent IFN production. Suppression of the cGAS-STING pathway (by siRNA) led to lack of DNA sensing, inhibition of IFN expression, and restoration of RNA virus replication.

The paper is very concisely and succinctly written and great fun to read. However, in this reviewer's view, more attention could be given to the clarification of the apparently very multifactorial functions of the cohesin complex. In lines 229-250 this could be pursued by the addition of a diagram in which the reaction cascades of innate immunity are linked to RVA and other RNA virus infections on one side and cohesin activities on the other side. Since STAG2^{-/-} cells show DNA damage consisting of an increase of genomic, non-mitochondrial DNA found in the cytoplasm, the question arises to what extent normal cellular functions were maintained and whether it was due to 'cell sickness' that less RVA was replicated. In this context, further explanation of the effect of cohesin suppression on multiple steps of RVA replication would be helpful. In a broader context it should be considered that cohesin has been shown to repair cellular DNA damage and that inversely cohesin suppression may make cellular DNA damage more apparent. Lack of cohesin and of its normal functions may lead to aneuploidy/cancerogenesis, and several severe genetic diseases have been classified as 'cohesinopathies'.

Specific Comments (requiring attention)

Line

76 Dataset S1 is very complex. It is suggested to produce a sub-Table for the main Text showing the data for the genes mentioned and also those for some genes not found to be critical for RV replication. The individual columns of the dataset should be explained in a Legend.

93f More details should be provided on the different STAG2-deleted cell lines. How 'sick' were they? Did they show cell division, and could they be passaged? As noted in General Comments, STAG2 deletion may lead to 'cohesinopathies' with very damaged cells and severe clinical symptoms e.g.,

Banerji R, Skibbens RV, Iovine MK. How many roads lead to cohesinopathies? *Dev Dyn*. 2017 Apr 19. doi: 10.1002/dvdy.24510. [Epub ahead of print] Review.

Mullegama SV, Klein SD, Mulatinho MV, Senaratne TN, Singh K; UCLA Clinical Genomics Center, Nguyen DC, Gallant NM, Strom SP, Ghahremani S, Rao NP, Martinez-Agosto JA. De novo loss-of-function variants in STAG2 are associated with developmental delay, microcephaly, and congenital anomalies. *Am J Med Genet A*. 2017 May;173(5):1319-1327.

Muto A, Schilling TF. Zebrafish as a model to study cohesin and cohesinopathies. *Methods Mol Biol*. 2017;1515:177-196.

Cucco F, Musio A. Genome stability: What we have learned from cohesinopathies. *Am J Med Genet C Semin Med Genet*. 2016 Jun;172(2):171-8.

Xu B, Lu S, Gerton JL. Roberts syndrome: A deficit in acetylated cohesin leads to nucleolar dysfunction. *Rare Dis.* 2014 Jan 21;2:e27743.

110 Consider whether the findings of Fig. S2 could suggest that STAG^{-/-} cells not only have a raised ISG and IFN response but also could be damaged to an extent that energetically they cannot support RVA replication.

111 Fig. S3c. It is not clear why the replication of several RNA viruses (flaviviruses) and of DNA viruses was not affected in STAG2^{-/-} cells.

122 Dataset S2. Please consider comments which are analogous to those made above on Dataset S1 (line 76).

140 The data of Fig. 2e support the notion that suppression of the STAT pathway in STAG2^{-/-} restored RVA and VSV replication via the mechanism investigated, but do not say more about the longer term viability of STAG2^{-/-} cells. Such data should be provided.

185 The observation that STAG2^{-/-}-STING^{-/-} cells were susceptible to RVA replication confirms the data of line 140. [See comment above.]. However, the observation that these cells appeared to have a decrease in double-stranded DNA breaks (DSBs) requires further exploration: did this double defect make the cell less ill, and how?

239f to line 250. The comments on the significance of the findings of this report on cancer biology should be reconsidered and either be omitted or rephrased to suggest how the data reported could be taken further in studies on cancer cells.

Specific Comments (relatively minor)

Line

1 Authors may reconsider the Title, depending on whether the main message is thought to be the inhibition of RNA virus replication by cohesin blockage or cellular pathophysiology in the absence of cohesin complex with blockage of replication of some RNA viruses as a secondary effect. At present, the Title reflects the second option. One could also formulate: 'Rotavirus replication is decreased by inhibition of the cellular cohesin complex with concomitant increase of the innate immune response', or similar.

50 Consider phrasing: ... including dengue virus, Zika virus, West Nile virus, hepatitis C virus , HIV...

56 Consider citation of: Saxena K, Simon LM, Zeng XL, Blutt SE, Crawford SE, Sastri NP, Karandikar UC, Ajami NJ, Zachos NC, Kovbasnjuk O, Donowitz M, Conner ME, Shaw CA, Estes MK. A paradox of transcriptional and functional innate interferon responses of human intestinal enteroids to enteric virus infection. *Proc Natl Acad Sci U S A.* 2017 Jan 24;114(4):E570-E579.

70 Add P and G genotypes of NCDV used. [See comment below.]

78 Consider citation of: Cheung W, Gill M, Esposito A, Kaminski CF, Courousse N, Chwetzoff S, Trugnan G, Keshavan N, Lever A, Desselberger U. Rotaviruses associate with cellular lipid droplet components to replicate in viroplasm, and compounds disrupting or blocking lipid droplets inhibit viroplasm formation and viral replication. *J Virol.* 2010 Jul;84(13):6782-98; Gaunt ER, Cheung W, Richards JE, Lever A, Desselberger U. Inhibition of rotavirus replication by downregulation of fatty acid synthesis. *J Gen Virol.* 2013 Jun;94(Pt 6):1310-7.

99 Fig. 1c, left panel. Clarify that 'mock' relates to mock transfection.

107 Extended Data Fig. 3a. According to this figure, the reduction of infectious progeny in STAG2^{-/-} cells is not significant for RV strains ST3 and 116E. The P and G types for all RVA strains used should be indicated in Methods. It is also of interest that the ratio of ffu/10⁴ cells differs substantially (by 1-2 log steps) between RVA strains.

195 ... we found that the HEK295 cells...

229 It is suggested to omit '... for the first time...'.

235 and 238. ... autoimmune-like manifestation... preventing autoimmune or lupus-like syndromes... Consider rephrasing for clarification and adding a reference for autoimmunity as cause of lupus (SLE).

314 ... and viral NSP5 level and virus yield were determined...

329 Fig. S1c. Do the nucleotide sequences shown span the whole exon 13? [In analogy, the same question relates to exon21 in Fig. S4e, exon3 in Fig. S7b, and exon 14 in Fig. S8a.]

497 Provide P and G types for all RVA strains used: NCDV, Wa, DS1, ST3, L26, WI61, 116E, 64M, RRV, SA11, UK, OSU, SB1A, ETD. [See comments above. This information may be important, since not all RVA strains are equally inhibited in STAG2^{-/-} cells (Fig. S3a).]

Reviewer #3 (Remarks to the Author):

Manuscript by Ding and coworkers reports that the loss-of-function of STAG2, member of the cohesin complex, confers human cell resistance to various rotaviruses. The authors involved

interferon signaling as the active pathway that suppresses viral infection in STAG2^{-/-} cells. Ding et al further identified increased basal DSB DNA damage response in STAG2^{-/-} cells, which ultimately led to an increase of cytosolic DNA that triggered IFN production via DNA sensing pathway by cGAS-STING.

Altogether this work identifies human genes that regulates viral infection and reveals unanticipated consequence of STAG2, and possibly cohesin itself, in susceptibility to RV infection.

Overall, the manuscript is well written and flows in logical manner. All experiments are properly performed, controlled and presented, and fully support authors' conclusions in most instances (see comments). I therefore believe this work should be considered further for publication in Nature Communications once concerns listed below would have been addressed.

Major comments:

I however have one important concern: though the involvement of SA2 loss-of-function in RV infection resistance through cytosolic DNA sounds clearly novel, I wonder whether such effect of SA2 inactivation is not the mere indirect consequence of increased basal DNA damage, which has been extensively reported for cohesin complex. In other word the authors should address the effect of increased basal DNA damage on RV's ability to infect cells. Would it differ from STAG2 inactivation?

Second, the authors extensively and unambiguously described consequences of STAG2 on RV infection; as STAG2 is member of cohesin, it would be essential to address whether cohesin inactivation recapitulates that of STAG2. This could be addressed by performing key experiments using SA1 or, best, SMC3 (#209).

Finally, I want to stress that the CRISPR-mediated genome-wide inactivation approach, as performed and presented by the authors is a beautiful one.

Minor comments:

A little graphical sketch would nicely illustrate presented findings.

Line 34: significance should be replaced by significant

Line 120: ISG should be defined (IFN-stimulated genes?)

Reviewer #1:

Q: This elegant and well-written paper by the Greenberg and Carette laboratories performed a genome-wide CRISPR screen to identify host factors required for rotavirus (RV) infection. While they identified a number of genes known to be critical for RV infection, they also identified STAG2, a component of the cohesion complex, as a 'hit' whereby its gene editing/deletion resulted in diminished RV infection as well reduction of infection of a number of related and unrelated RNA viruses. Through a series of well-designed, well-controlled, and detailed studies, they identified the mechanism of action: a deficiency of STAG2 results in spontaneous DNA damage, accumulation of DNA in the cytosol, activation of the cGAS-STING pathway, and spontaneous production of type I IFNs. This work, which was initially designed to identify RV-dependent host factors, instead defines a new regulator of IFN homeostasis and host pathway (cohesion complex members) that could have implications for autoimmune diseases (e.g., lupus, interferonopathies), cancer, and cancer therapies that modulate cell-intrinsic immunity. The strengths of the manuscript include its approach, mechanistic depth, novelty of findings, and clarity of the data. This will be an important contribution to the field of immunity and autoimmunity. There are only few minor suggestions for improvement.

A: We appreciate the reviewer's complimentary summary of our work.

Q: Given that a deficiency of STAG2 results in spontaneous STING activation and IFN production, is it surprising that some but not all viruses are inhibited. In particular, DNA viruses, flaviviruses, and picornaviruses are not inhibited whereas rotavirus, alphaviruses (CHIKV), influenza, and VSV are inhibited. As this seems puzzling, the authors need to address this at least in the Discussion. What could be the explanation for why a robust IFN signature would inhibit some viruses but not others, as it seems likely that IFN pre-treatment would inhibit infection of all of these viruses. Related to this, in Figure 2, the authors show ISG and IFN- β induction by qRT-PCR and IFN- λ accumulation by ELISA in STAG2^{-/-} cells. Did they also directly measure secreted levels of IFN- α/β by ELISA using commercially available kits? Could a difference in induction levels of type I and III IFNs explain the disparity in effects against the different viruses?

A: We thank the reviewer for these very thoughtful comments. Indeed, we were initially puzzled as well when we observed a differential inhibition of rotavirus, CHIKV, VSV and influenza in contrast to DNA viruses, flavivirus and picornavirus (Fig. 2A and Fig. S3C). We concur with the reviewer that this difference could be due to an IFN response skewed towards IFN- λ instead of IFN- α/β . We performed ELISA to measure IFN secretion in HT-29 cells as suggested by the reviewer and found that only IFN- λ (but not IFN- α , IFN- β or TNF- α) was significantly up-regulated in the supernatant of *STAG2*^{-/-} HT-29 cells (new Fig. S4C). Also, in another manuscript under submission, we found that this preferential induction of IFN- λ in response to RNA PAMPs is a unique trait of intestinal epithelial cells (Ding and Greenberg, unpublished data). We have now added this new experimental data (new Fig. S4C) and discussion of these findings (lines 118-120) to further strengthen our conclusions. Please note that all the line numbers referenced here correspond to those in the “Revised *STAG2* manuscript”..

Q: In all main and Supplemental Figure legends, the authors should indicate clearly the number of independent biological experiments, the number of technical replicates, and the statistical test used to determine significance. Related to this point, in the Methods, the authors state they used a Student's *t* test for all comparisons. However, the data in Fig 1c, 2e, 3c, S2a, S6a, and S7c requires an ANOVA with a multiple comparison's correction, as they compare across multiple groups.

A: We thank the reviewer for the helpful suggestion that the figure legends should include the number of biological experiments and replicates. These have now been added to all the main and supplemental figures. With regard to the means to determine statistical significance, we agree with the reviewer's opinion and have recalculated the statistics of all column data. In the revised **Statistical Analysis** section, statistical significance in Figs. 2A, 2C, S1B, S2D, S3A, S3C, S4E, S5A, S5D, S6B was calculated by Student's *t* test and statistical significance in Figs. 1C, 2D, 2E, 3C, S1E, S2A, S4B, S4C, S5E, S6A, S7C, S8E was calculated by a pairwise ANOVA test using Prism 7.0c (GraphPad). Statistical significance of Fig. S2B, where the synthesis of positive and negative strands of rotavirus RNA were compared between WT and *STAG2*^{-/-} cells over the time course, was calculated by Two-Way ANOVA test. All of changes have been clarified in the Supplement lines 154-162.

Q: Summary. The authors started the Summary with a few sentences on the cohesion complex. Given why the screen was performed (to identify host factors required for RV infection), they should re-order the text to reflect the true sequence of events in the paper. The statement “whether cohesion participates in regulation of innate immune responses...” seems disingenuous to this Reviewer – prior to this study, this was not even a consideration.

A: We thank the reviewer for the suggestion. We have now rephrased the summary to reflect the original rationale of our screening approach (lines 27-29).

Q: Human mutations in *STAG2*. Are there known mutations in *STAG2* that have been detected in cancer patients (would seem so based on references 3 and 32)? Do these have any association with autoimmune phenotypes? Did the authors test any of the point mutations for loss-of-function vis a vis regulation of IFN responses?

A: We thank the reviewer for raising this important point for clarification. Actually, the premature stop codon mutations that we examined in Fig. S9B were originally identified

in cancer patients (Kim JS, PLoS Genetics, 2016). In fact, approximately 85% of tumor-derived *STAG2* mutations lead to premature truncation of the encoded protein. In response to the reviewer's suggestion, we now provide additional molecular details on the role of *STAG2* point mutations in regulating DNA damage and the IFN signaling. We cloned Gateway DONOR plasmid expressing WT and a series of point mutations in *STAG2*: E134D, R252Q, R541K, L997W (Kim E, Cancer Discovery, 2016) into the pG-LAP6 destination vector that expressed an N-terminal GFP tag (Ding S, PLoS Pathogens, 2016). We then transfected *STAG2*^{-/-} HEK293 cells reconstituted with an intact cGAS-STING pathway with these constructs and examined STAT1 phosphorylation, indicative of IFN activation. As is shown below, we found that with comparable levels of *STAG2* expression, only WT *STAG2* was able to restore the IFN homeostasis, suggesting that these point mutations could potentially contribute to autoimmune phenotypes by promoting an excess production of IFN. This new data is now discussed in lines 197-200 of the revised manuscript.

Q: Figure S4c. Why is the IRF3 staining so dim in the mock-treated WT cells relative to the mock-treated *STAG2*^{-/-} cells? Usually, IRF3 is expressed strongly at the basal level. Was it one of the induced genes in these cells by their RNAseq analysis?

A: We agree with the reviewer that IRF3 is normally expressed at a high baseline level. However, because of its diffuse cytoplasmic localization, the immunofluorescence signal of IRF3 is not as strong and concentrated as that post nuclear translocation. IRF7 and IRF9 were up-regulated 6-fold and 11-fold respectively in *STAG2*^{-/-} HT-29 cells compared to the WT HT-29 cells (Dataset S2). However, none of the other IRFs, including IRF3, was affected by the loss of *STAG2*.

Q: The authors state a nuclear source of the DSB and cytosolic DNA in *STAG2*^{-/-} cells. How did they rule out cleaved mitochondrial DNA as a source?

A: We apologize for our lack of precision in communication in this part of the manuscript. We concur that the nature and identity of the cytoplasmic DNA responsible for activating the cGAS-STING DNA sensing pathway in *STAG2*^{-/-} cells is an important topic for analysis. During the review process of our manuscript, four studies were published in sequence (Dou Z, Nature, 2017; Harding SM, Nature, 2017; Mackenzie KJ, Nature, 2017; Gluck S, Nature Cell Biology, 2017), reporting a similar finding: cytoplasmic DNA PAMP stemming from genomic instability gives rise to IFN activation through the DNA sensing pathway. Careful scrutiny by these researchers leads to the conclusion that the PAMP is of nuclear origin from DNA damage response induced by senescence or oxidizing agent. We have now clarified this point and discussed these results in the manuscript (lines 225-228).

Q: Line 127. Should read “secretion of IFN- λ ”.

A: We have made the correction as suggested in line 117.

Q: Line 182 and 186. The authors state that STAT1 hyper-phosphorylation was abolished and that there was a decrease in DSBs in STAG2^{-/-}-STING^{-/-} cells in Fig S7b. This Figure panel does not show either. Is the data elsewhere?

A: We apologize for the possible confusion. The decrease in DSBs in STAG2^{-/-}STING^{-/-} cells as compared to STAG2^{-/-} cells is shown as reduced γ H2AX levels in the western blot (4th row of Fig. S7B), leading us to reason that the IFN signaling may reinforce the DNA damage in host cells.

Q: Line 198. Perhaps the authors should state “naturally lack cGAS or STING expression”? See Lane 3 of Fig S8c.

A: We have made the correction in line 186.

Q: Line 218 and Fig 4a. The authors state there was a complete KO of STAG2 in 30% of the primary IECs. Yet in the Figure (panels b-d), they refer to this as a complete KO. This seems confusing. Related to this, in the legend they state that “bar graphs in b-d are calculated based on comparison between WT and STAG2 mosaics. What bar graphs? (scatter plots are shown). Did the authors normalize the data in some way – this section needs to be clarified.

A: We thank the reviewer for pointing out the inconsistency. The human intestinal enteroids in Fig. 4 were a mosaic composed of both WT and STAG2^{-/-} IECs. 30% KO efficiency is calculated based on the quantification of immunofluorescence images of at least 10 independent enteroids post transduction. We have clarified the Fig. 4 legend by stating these are “partial STAG2^{-/-} ileum enteroids”. We also changed “bar graphs” to “scatter plots” as suggested (lines 311-314).

Q: Lines 247-250. This conclusion seems obvious. cGAS and STING ligands will not work in cells lacking these pathways. Perhaps the authors can articulate a more interesting coda to this elegant paper?

A: We have now added more discussion (lines 225-228, 238-241) to the manuscript, in particular with regard to the recent publications on a similar topic, as mentioned above, and on the likely futility of oncolytic viral therapy on STAG KO cancers.

Reviewer #2:

Q: This manuscript describes the application of a recently established genome-wide CRISPR/Cas9 screening procedure to the identification of cellular genes important for the replication of species A rotaviruses (RVAs) in different gut-derived human cell lines and also in human gut stem cell-derived intestinal 3D (enteroid) cultures. Genes known to be critical for RVA replication were confirmed, e.g. genes involved in sialic acid and glycosphingolipid biosynthesis, and the gene encoding a fatty acid 2-hydroxylase. In addition the very interesting and novel observation was made, consisting of showing that suppression of the cellular cohesin gene STAG2, encoding stromal antigen 2, or of other genes of the cohesin complex confers significant decrease of RVA replication. This was shown in different cell lines including enteroids

and with different RVA strains. The finding is also interesting since the cohesin complex functions in the nucleus while RVA replication takes place in the cytoplasm. The causative role of cohesin genes was supported by exogenous expression of STAG2 in STAG2^{-/-} cells which restored the ability of RVA to replicate. The replication of several other RNA viruses, such as vesicular stomatitis virus, chikungunya virus and different influenza A virus subtypes, was also inhibited in STAG2^{-/-} cells, but not of some DNA viruses. It turned out that in uninfected STAG2^{-/-} cells interferon (IFN) stimulatory genes (ISGs) such as encoding MX1, IFITM1, IFI6, etc were significantly upregulated, explaining the inhibitory effect on RNA virus replication. Mechanistically, it was found that STAG2^{-/-} cells exhibit signs of intracellular DNA damage, leading to activation of the cGAS-STING pathway of DNA sensing and subsequent IFN production. Suppression of the cGAS-STING pathway (by siRNA) led to lack of DNA sensing, inhibition of IFN expression, and restoration of RNA virus replication.

The paper is very concisely and succinctly written and great fun to read. However, in this reviewer's view, more attention could be given to the clarification of the apparently very multifactorial functions of the cohesin complex. In lines 229-250 this could be pursued by the addition of a diagram in which the reaction cascades of innate immunity are linked to RVA and other RNA virus infections on one side and cohesin activities on the other side. Since STAG2^{-/-} cells show DNA damage consisting of an increase of genomic, non-mitochondrial DNA found in the cytoplasm, the question arises to what extent normal cellular functions were maintained and whether it was due to 'cell sickness' that less RVA was replicated. In this context, further explanation of the effect of cohesin suppression on multiple steps of RVA replication would be helpful. In a broader context it should be considered that cohesin has been shown to repair cellular DNA damage and that inversely cohesin suppression may make cellular DNA damage more apparent. Lack of cohesin and of its normal functions may lead to aneuploidy/cancerogenesis, and several severe genetic diseases have been classified as 'cohesinopathies'.

A: We appreciate the reviewer's detailed and positive summary of our work.

Q: Line 76 Dataset S1 is very complex. It is suggested to produce a sub-Table for the main Text showing the data for the genes mentioned and also those for some genes not found to be critical for RV replication. The individual columns of the dataset should be explained in a Legend.

A: We thank the reviewer for the suggestion. We have now highlighted the genes mentioned in the manuscript in yellow and provided additional detailed explanation to each column in the Dataset S1 legend (Supplement lines 326-332).

Q: 93f More details should be provided on the different STAG2-deleted cell lines. How 'sick' were they? Did they show cell division, and could they be passaged?

As noted in General Comments, STAG2 deletion may lead to 'cohesinopathies' with very damaged cells and severe clinical symptoms e.g.,

Banerji R, Skibbens RV, Iovine MK. How many roads lead to cohesinopathies?

Dev Dyn. 2017 Apr 19. doi: 10.1002/dvdy.24510. [Epub ahead of print] Review. Mullegama SV, Klein SD, Mulatinho MV, Senaratne TN, Singh K; UCLA Clinical Genomics Center,

Nguyen DC, Gallant NM, Strom SP, Ghahremani S, Rao NP, Martinez-Agosto JA. De novo loss-of-function variants in STAG2 are associated with developmental delay, microcephaly, and congenital anomalies. Am J Med Genet A. 2017 May;173(5):1319-1327.

Muto A, Schilling TF. Zebrafish as a model to study cohesin and cohesinopathies. *Methods Mol Biol.* 2017;1515:177-196.

Cucco F, Musio A. Genome stability: What we have learned from cohesinopathies. *Am J Med Genet C Semin Med Genet.* 2016 Jun;172(2):171-8.

Xu B, Lu S, Gerton JL. Roberts syndrome: A deficit in acetylated cohesin leads to nucleolar dysfunction. *Rare Dis.* 2014 Jan 21;2:e27743.

A: The reviewer raises several important questions that we have now further investigated. In brief, all of our STAG2 deleted cells do not appear “sick” and can be passaged normally, which is one of the prerequisites of generating CRISPR-Cas9 knockout cells. To measure the cytotoxicity of STAG2 deficiency, we performed trypan blue staining directly using cells grown in 6-well plates as well as live/dead staining by flow cytometry. Both assays revealed that despite a slightly higher percentage of dead cells within the *STAG2*^{-/-} population, they do not present a severe defect in survival compared to WT and *STAG2*^{-/-}*STING*^{-/-} cells (new Figs. S1D and E). In these assays, we used STS (staurosporine, apoptosis inducer) treatment and STS plus Z-VAD (Z-VAD-FMK, apoptosis inhibitor) on WT cells as positive and negative controls respectively. In addition, we also measured cell proliferation in WT, *STAG2*^{-/-}, and *STAG2*^{-/-}*STING*^{-/-} cells using a metabolism-based luminescence test. We found that within the 48-hour period post seeding, STAG2 deficiency did not cause retardation in cell growth (new Fig. S1F). Taken together, these new data support the conclusion that STAG2 is not critical for cell survival and proliferation in cell culture.

However, we would like to add that, in collaboration with Dr. David Solomon at UCSF, we have recently obtained complete *Stag2* knockout mice (western blot and RT-qPCR on intestinal tissues, see below), which were born at a sub-Mendelian ratio, suggesting that there are potential developmental defects associated with the loss of *Stag2* in the whole organism.

Q: 110 Consider whether the findings of Fig. S2 could suggest that *STAG2*^{-/-} cells not only have a raised ISG and IFN response but also could be damaged to an extent that energetically they cannot support RVA replication.

A: We thank the reviewer for this interesting comment. While the data in *STAG2*^{-/-} cells may give rise to other possibilities of RVA replication inhibition, we found that both *STAG2*^{-/-}*STING*^{-/-} cells and *STAG2*^{-/-}*STING*^{-/-} cells, which have normal levels of IFN and ISG expression, support RVA replication (Figs. 2E and S4F). Thus, these data led us to conclude that the induction of IFN and ISGs is the primary mechanism of RVA inhibition.

Q: 111 Fig. S3c. It is not clear why the replication of several RNA viruses (flaviviruses) and of DNA viruses was not affected in *STAG2*^{-/-} cells.

A: We thank the reviewer for raising this important point, which was also brought up by reviewer #1. As described above, our *STAG2*^{-/-} HT-29 cells predominantly produce more type III IFN (IFN- λ) than type I (IFN- α/β) (Figs. 2C, 2D and new S4C). Although it is generally considered that IFN- λ and IFN- α/β induce the same JAK-STAT signaling and overlapping ISG expression, viruses from distinct families may respond differently to these IFNs, i.e. possibly weak suppression of flavivirus replication by IFN- λ , which acts at mucosal surfaces *in vivo*. This hypothesis will be examined in future studies.

Q: 122 Dataset S2. Please consider comments which are analogous to those made above on Dataset S1 (line 76).

A: We have added more description to Dataset S2 as suggested (Supplement lines 335-345).

Q: 140 The data of Fig. 2e support the notion that suppression of the STAT pathway in *STAG2*^{-/-} restored RVA and VSV replication via the mechanism investigated, but do not say more about the longer term viability of *STAG2*^{-/-} cells. Such data should be provided.

Q: 185 The observation that *STAG2*^{-/-}*STING*^{-/-} cells were susceptible to RVA replication confirms the data of line 140. [See comment above.]. However, the observation that these cells

appeared to have a decrease in double-stranded DNA breaks (DSBs) requires further exploration: did this double defect make the cell less ill, and how?

A: As noted above, we have carefully assessed the viability of WT, STAG2^{-/-} and STAG2^{-/-} STING^{-/-} cells and did not find a significant inhibition of growth due to the loss of STAG2 (new Figs. S1D-F).

Q: 239f to line 250. The comments on the significance of the findings of this report on cancer biology should be reconsidered and either be omitted or rephrased to suggest how the data reported could be taken further in studies on cancer cells.

A: We thank the reviewer for this comment and have added more discussion as to why our finding might be important for future cancer studies (see lines 238-241).

Q: Line 1 Authors may reconsider the Title, depending on whether the main message is thought to be the inhibition of RNA virus replication by cohesin blockage or cellular pathophysiology in the absence of cohesin complex with blockage of replication of some RNA viruses as a secondary effect. At present, the Title reflects the second option. One could also formulate: 'Rotavirus replication is decreased by inhibition of the cellular cohesin complex with concomitant increase of the innate immune response', or similar.

A: Our title indeed reflects the second option as intended.

Q: 50 Consider phrasing: ... including dengue virus, Zika virus, West Nile virus, hepatitis C virus, HIV...

A: We have made the correction as suggested (line 45).

Q: 56 Consider citation of: Saxena K, Simon LM, Zeng XL, Blutt SE, Crawford SE, Sastri NP, Karandikar UC, Ajami NJ, Zachos NC, Kovbasnjuk O, Donowitz M, Conner ME, Shaw CA, Estes MK. A paradox of transcriptional and functional innate interferon responses of human intestinal enteroids to enteric virus infection. Proc Natl Acad Sci U S A. 2017 Jan 24;114(4):E570-E579.

A: We have cited the Saxena K paper as Ref 28 in line 204.

Q: 70 Add P and G genotypes of NCDV used. [See comment below.]

A: We have added the genotype information of all the RV strains used in this study to the legend of Fig. S3 (Supplement lines 203-205).

Q: 78 Consider citation of: Cheung W, Gill M, Esposito A, Kaminski CF, Courousse N, Chwetzoff S, Trugnan G, Keshavan N, Lever A, Desselberger U. Rotaviruses associate with cellular lipid droplet components to replicate in viroplasms, and compounds disrupting or blocking lipid droplets inhibit viroplasm formation and viral replication. J Virol. 2010 Jul;84(13):6782-98; Gaunt ER, Cheung W, Richards JE, Lever A, Desselberger U. Inhibition of rotavirus replication by downregulation of fatty acid synthesis. J Gen Virol. 2013 Jun;94(Pt 6):1310-7.

A: As suggested, we have cited the Gaunt ER paper as Ref 16 in line 68.

Q: 99 Fig. 1c, left panel. Clarify that 'mock' relates to mock transfection. 107 Extended Data Fig. 3a. According to this figure, the reduction of infectious progeny in STAG2^{-/-} cells is not significant for RV strains ST3 and 116E. The P and G types for all RVA strains used should be indicated in Methods. It is also of interest that the ratio of ffu/104 cells differs substantially (by 1-2 log steps) between RVA strains.

A: As noted above, we have added the RV genotype information as suggested.

Q: 195 ... we found that the HEK295 cells...

A: We have changed "HEK295" to "HEK293".

Q: 229 It is suggested to omit '... for the first time...'.

A: We have left out "for the first time" as suggested.

Q: 235 and 238. ... autoimmune-like manifestation... preventing autoimmune or lupus-like syndromes... Consider rephrasing for clarification and adding a reference for autoimmunity as cause of lupus (SLE).

A: As the reviewer has suggested, we now provided a citation (Ref 33) linking autoimmunity (excessive IFN production) to SLE (Banchereau J, Immunity, 2006; line 224).

Q: 314 ... and viral NSP5 level and virus yield were determined...

A: We thank the reviewer for pointing out this grammatical error. We have now changed "was" to "were" (line 333).

Q: 329 Fig. S1c. Do the nucleotide sequences shown span the whole exon 13? [In analogy, the same question relates to exon21 in Fig. S4e, exon3 in Fig. S7b, and exon 14 in Fig. S8a.]

A: We apologize for the possible confusion. The nucleotide sequences provided only cover a small region within each exon of all three genes (STAG2, STAT1 and TMEM173). We have moved the labeling bars to more truthfully reflect the gene editing region (Figs. S1C, S4F, S7B).

Q: 497 Provide P and G types for all RVA strains used: NCDV, Wa, DS1, ST3, L26, WI61, 116E, 64M, RRV, SA11, UK, OSU, SB1A, ETD. [See comments above. This information may be important, since not all RVA strains are equally inhibited in STAG2^{-/-} cells (Fig. S3a).]

A: We agree with the reviewer that genotype information is important and is also the reason behind the wide span of RV strains that we tested in this study. We have now provided this information, as suggested, to the figure legend of Fig. S3A (Supplement lines 203-205).

Reviewer #3:

Q: Manuscript by Ding and coworkers reports that the loss-of-function of STAG2, member of the cohesin complex, confers human cell resistance to various rotaviruses. The authors involved interferon signaling as the active pathway that suppresses viral infection in STAG2^{-/-} cells. Ding

et al further identified increased basal DSB DNA damage response in STAG2 ^{-/-} cells, which ultimately led to an increase of cytosolic DNA that triggered IFN production via DNA sensing pathway by cGAS-STING.

Altogether this work identifies human genes that regulates viral infection and reveals unanticipated consequence of STAG2, and possibly cohesin itself, in susceptibility to RV infection. Overall, the manuscript is well written and flows in logical manner. All experiments are properly performed, controlled and presented, and fully support authors' conclusions in most instances (see comments). I therefore believe this work should be considered further for publication in Nature Communications once concerns listed below would have been addressed.

A: We appreciate the favorable comments from the reviewer and have provided additional information to strengthen our manuscript.

Q: I however have one important concern: though the involvement of SA2 loss-of-function in RV infection resistance through cytosolic DNA sounds clearly novel, I wonder whether such effect of SA2 inactivation is not the mere indirect consequence of increased basal DNA damage, which has been extensively reported for cohesin complex. In other word the authors should address the effect of increased basal DNA damage on RV's ability to infect cells. Would it differ from STAG2 inactivation?

Second, the authors extensively and unambiguously described consequences of STAG2 on RV infection; as STAG2 is member of cohesin, it would be essential to address whether cohesin inactivation recapitulates that of STAG2. This could be addressed by performing key experiments using SA1 or, best, SMC3 (#209).

Finally, I want to stress that the CRISPR-mediated genome-wide inactivation approach, as performed and presented by the authors is a beautiful one.

A: We are thankful to these great suggestions and agree with the reviewer that the role of DNA damage itself in RV replication is worth investigation. To address the first question: we have examined the effect of increased basal DNA damage on RV replication. Our preliminary data suggest that treatment of WT HT-29 cells with etoposide (ETP), a topoisomerase II inhibitor that triggers a DNA damage signaling, mirrors STAG2 deficiency and induces STAT1 phosphorylation and IFN and ISG expression (new Figs. S5C and D). Consistently, we observed a decrease in intracellular RV RNA levels in ETP-treated cells (new Fig. 5E). Taken together, these data (lines 151-154) indicate that independent of the STAG2 status, elevated genomic DNA damage is sufficient to activate an IFN response and inhibit RV replication.

To answer the second question: although we did not attempt to produce *SMC3* knockout cells, we successfully generated HT-29 cells lacking *PAXIP1* (#30), a gene that pairs with *PAGR1* (#31) and functions in the DNA damage repair pathways (Callen E, Cell, 2013). In accordance with our data on *STAG2*, *PAXIP1*^{-/-} HT-29 cells have hyper *STAT1* phosphorylation and enhanced expression levels of IFN-λ and MX1 prior to any virus infections (see below). In addition, *PAXIP1*^{-/-} cells completely phenocopy *STAG2*^{-/-} cells and are resistant to infection by a panel of RNA viruses, including RV, CHIKV, and VSV, but remain susceptible to human rhinovirus (HRV-A2) infection (see below). These new results suggest that the cohesin complex subunits, as well as other components in the DNA damage repair signaling, are important regulators of IFN homeostasis.

Q: A little graphical sketch would nicely illustrate presented findings.

A: We agree with the reviewer and have now added an illustration/graphical abstract to summarize our findings (new Fig. S10, lines 216-218, Supplement lines 297-304).

Q: Line 34: significance should be replaced by significant

A: We apologize for the typo and have corrected as suggested (line 30).

Q: Line 120: ISG should be defined (IFN-stimulated genes?)

A: We have now added IFN-stimulated genes to define ISGs as it first appears in the main text (line 111).

REVIEWERS' COMMENTS:

Reviewer #2 (Remarks to the Author):

STAG2 deficiency induces interferon responses via cGAS-STING pathway and restricts virus infection

By Siyuan Ding et al (Corresponding author: Harry B Greenberg)

Submitted to Nat Commun (Editorial No. NCOMMS-17-19147A)

General Comments

This is the revised version of a manuscript the original of which has been studied and commented upon by this reviewer. The authors have considered the comments and suggestions of three reviewers very carefully, clarified various issues and carried out additional experiments to strengthen the manuscript. The replies to reviewers are of exemplary clarity and comprehensiveness. The revised manuscript is a substantial improvement of an already remarkable original submission.

This reviewer found their questions/suggestions overwhelmingly answered and dealt with in a very satisfactory way. Only some rather minor additional Specific Comments remain which are listed below.

Specific Comments

A. Main text

Line

25 Read: ... that coordinates

31 Consider phrasing: ... intestinal enteroids and also to the replication of other RNA viruses.
[This will explain that RVs are not specifically mentioned in the Title.]

33 ... pathway. The resulting activation...

64 ... until the appearance of visibly apparent surviving colonies...

106 ... and influenza viruses.

118 ... Fig. 2C and 2D...

142 ... HT-29 cells than in WT cells...

184 ... we found that the HEK293 cells...

206 ... of STAG2 at the protein level...

216 Omit: '...for the first time...'

218 ... The genome-wide loss-of-function...

274 ... For (c) experiments were... [There is no panel (d) of Fig. 1.]

320 The complete ref. is: Nat Rev Microbiol. 2017 Jun;15(6):351-364.

326 The complete ref. is: Nat Genet. 2017 Feb;49(2):193-203.

331 The ref. should read: Estes MK, Greenberg HB. Rotaviruses. In: Knipe DM, Howley PM, et al., editors. Fields Virology, 6th ed. p. 1347–1401. Philadelphia: Wolters Kluwer Health/Lippincott Williams & Wilkins; 2013.

378 Ref. 24 is incomplete.

381 Ref. 25 is incomplete.

401 Ref. 33 adjust to line.

418 The complete ref. is: Nat Nanotechnol. 2017 Jul;12(7):648-654.

B. Supplementary Materials

68 Transfection of purified RV DLPs is used for transfection (Fig. S2). The method (classical) should be referred to by appropriate refs.

185f See comment line 68.

194 ... standard plaque assay. Refer to ref. 4.

216f Fig. S4. Panel A left duplicates panel B of Fig. 2. Panel A right should be slightly enlarged.

Reviewer #3 (Remarks to the Author):

This revised version has been largely improved by both significant rewriting of some parts of the manuscript and addition of complementary experiments that all together further strengthen and clarify its main message.

I fully support acceptance and publication of this work in Nature Communications

**Response to the editorial comments on the Nature Communications Manuscript
(Editorial No: NCOMMS-17-19147A)**

Ding S, *et al.*, **STAG2 deficiency induces IFN responses via cGAS-STING pathway and restricts virus infection**

Dear editor,

We appreciate the opportunity to re-submit our revised Nature Communications Manuscript NCOMMS-17-19147-A and would like to thank all reviewers for their supportive comments and thoughtful suggestions. We have now made further edits in our manuscript to improve clarity.

Below please find our point-by-point response to the reviewers' comments.

Reviewer #2:

General Comments

This is the revised version of a manuscript the original of which has been studied and commented upon by this reviewer. The authors have considered the comments and suggestions of three reviewers very carefully, clarified various issues and carried out additional experiments to strengthen the manuscript. The replies to reviewers are of exemplary clarity and comprehensiveness. The revised manuscript is a substantial improvement of an already remarkable original submission.

This reviewer found their questions/suggestions overwhelmingly answered and dealt with in a very satisfactory way. Only some rather minor additional Specific Comments remain which are listed below.

We appreciate the complimentary comments from the reviewer and we have further modified our manuscript to improve clarity.

Specific Comments

A. Main text

Line 25 Read: ... that coordinates

Corrected as suggested.

*31 Consider phrasing: ... intestinal enteroids and also to the replication of other RNA viruses.
[This will explain that RVs are not specifically mentioned in the Title.]*

We thank the reviewer for this thoughtful comment. Since rotavirus is the only pathogen examined in the enteroid system, we left this sentence as it is.

33 ... pathway. The resulting activation...

As adjectives, “resultant” and “resulting” have the same meaning, so we left the word as it is.

64 ... until the appearance of visibly apparent surviving colonies...

We have replaced “formation” with “appearance” as suggested.

106 ... and influenza viruses.

Corrected as suggested.

118 ... Fig. 2C and 2D...

Corrected as suggested.

142 ... HT-29 cells than in WT cells...

Corrected as suggested.

184 ... we found that the HEK293 cells...

Corrected as suggested.

206 ... of STAG2 at the protein level...

Corrected as suggested.

216 Omit: ‘...for the first time...’

Deleted as suggested.

218 ... The genome-wide loss-of-function...

Corrected as suggested.

274 ... For (c) experiments were... [There is no panel (d) of Fig. 1.]

Corrected as suggested.

320 The complete ref. is: Nat Rev Microbiol. 2017 Jun;15(6):351-364.

Corrected as suggested.

326 The complete ref. is: Nat Genet. 2017 Feb;49(2):193-203.

Corrected as suggested.

331 *The ref. should read: Estes MK, Greenberg HB. Rotaviruses. In: Knipe DM, Howley PM, et al., editors. Fields Virology, 6th ed. p. 1347–1401. Philadelphia: Wolters Kluwer Health/Lippincott Williams & Wilkins; 2013.*

Corrected as suggested.

378 *Ref. 24 is incomplete.*

Corrected as suggested.

381 *Ref. 25 is incomplete.*

Corrected as suggested.

401 *Ref. 33 adjust to line.*

Corrected as suggested.

418 *The complete ref. is: Nat Nanotechnol. 2017 Jul;12(7):648-654.*

Corrected as suggested.

B. Supplementary Materials

68 *Transfection of purified RV DLPs is used for transfection (Fig. S2). The method (classical) should be referred to by appropriate refs.*

A new reference (Bass DM, JCI, 1992) citing the use of DLPs has now been added to the supplementary information.

185f *See comment line 68.*

A new reference (Urasawa T, Micro and Immuno, 1981) has been added as suggested.

194 *... standard plaque assay. Refer to ref. 4.*

Corrected as suggested.

216f *Fig. S4. Panel A left duplicates panel B of Fig. 2. Panel A right should be slightly enlarged.*

Corrected as suggested.

Reviewer #3:

This revised version has been largely improved by both significant rewriting of some parts of the manuscript and addition of complementary experiments that all together further strengthen and clarify its main message.

I fully support acceptance and publication of this work in Nature Communications

We are grateful to the reviewer's laudatory comments.